# Landscape analysis of nutrition services at Primary Health Care Units (PHCUs) in four districts of Ethiopia

**Esete Habtemariam Fenta**[1]*, **Bilal Shikur Endris**[1], **Yalemwork Getenet Mengistu**[1], **Fekadu Elias Sadamo**[2], **Endashaw Hailu Gelan**[1], **Tsegaye Gebrezgher Beyene**[1], **Seifu Hagos Gebreyesus**[1]

1 Department of Nutrition and Dietetics, School of Public Health, College of Health Sciences, Addis Ababa University, Addis Ababa, Ethiopia, 2 Department of Reproductive Health and Nutrition, School of Public Health, College of Health Sciences, Wolaita Sodo University, Wolaita Sodo, Ethiopia

* esete.f@gmail.com

## Abstract

### Background

Good nutrition and healthy growth during the first 1000days have lasting benefit throughout life. For this, equally important is the structural readiness of health facilities. However, structural readiness and nutrition services provision during the first 1000 days in Ethiopia is not well understood. The present study was part of a broader implementation research aimed at developing model nutrition districts by implementing evidence based, high impact and cost-effective package of nutrition interventions through the continuum of care. This study was aimed at assessing structural readiness of health facilities and the extent of nutrition service provision in the implementation districts.

### Methods

This assessment was conducted in four districts of Ethiopia. We used mixed method; a quantitative study followed by qualitative exploration. The quantitative part of the study addressed two-dimensions, structural readiness and process of nutrition service delivery. The first dimension assessed attributes of context in which care is delivered by observing availability of essential logistics. The second dimension assessed the service provision through direct observation of care at different units of health facilities. For these dimensions, we conducted a total of 380 observations in 23 health centers and 33 health posts. The observations were conducted at the Integrated Management of Neonatal and Childhood Illnesses unit, immunization unit, Antenatal care unit and Postnatal care unit. The qualitative part included a total of 60 key informant interviews with key stakeholders and service providers.

### Result

We assessed structural readiness of 56 health facilities. Both quantitative and qualitative findings revealed poor structural readiness and gap in nutrition services provision. Health

**Data Availability Statement:** All relevant data are within the manuscript and its Supporting Information files.

**Funding:** SHG and BSE have received one grant from Irish Aid Ethiopia (https://www.irishaid.ie/what-we-do/countries-where-we-work/our-partner-countries/ethiopia/) through Ministry of Health, Ethiopia. The funders had no role in the study design, data collection and analysis, decision to publish, or preparation of the manuscript.

**Competing interests:** The authors have declared that no competing interests exist.

facilities lack essential logistics which was found to be more prominent at health posts compared to health centers. The process evaluation showed a critical missed opportunity for anthropometric assessment and preventive nutrition counselling at different contact points. This was particularly prominent at immunization unit (where only 16.4% of children had their weight measured and only 16.2% of mothers with children under six month of age were counselled about exclusive breastfeeding). Although 90.4% of pregnant women who came for antenatal care were prescribed iron and folic acid supplementation, only 57.7% were counselled about the benefit and 42.4% were counselled about the side effect. The qualitative findings showed major service provision bottlenecks including non-functionality of the existing district nutrition coordination body and technical committees, training gaps, staff shortage, high staff turnover resulting in work related burden, fatigue and poor motivation among service providers.

## Conclusion

We found a considerable poor structural readiness and gaps in delivering integrated nutrition services with a significant missed opportunity in nutrition screening and counselling. Ensuring availability of logistics and improving access to training might improve delivery of nutrition services. In addition, ensuring adequate human resource might reduce missed opportunity and enable providers to provide a thorough preventive counselling service.

## Background

Maternal and child undernutrition remain prevalent with important consequence in low-income and middle-income countries [1,2]. According to the 2018 World Health Organization (WHO) African region data, between 2000 and 2015 nine countries have prevalence rate of maternal underweight above 15% and exceeds 20% in Ethiopia [3]. Undernutrition is responsible for 45% of all under five child deaths, representing more than 3 million deaths per year [1]. Ethiopia has recorded remarkable improvements in health status and human centered development over the past two decades. Under five mortality rate decreased from 203 per 1000 live birth in 1990 to 58 per 1000 live birth in 2016 [4]. With respect to nutrition indicators, between 2000 and 2016, the national stunting prevalence rate declined by 20% from 58% to 38%, whilst underweight declined by 17% from 41% to 24%, and wasting from 12% to 10% [5].

Even though there is an improvement in undernutrition over the years, stunting, wasting and underweight remain to be prevalent and important public health problem in Ethiopia. In an effort to accelerate the reduction of undernutrition, the Government of Ethiopia developed national nutrition strategy and national nutrition programs (NNP). The revised NNP aims to strategically address nutrition problem by focusing on the life cycle approach to map key actions needed to improve the nutritional status of women and children starting with the first thousand days including nutrition actions for in and out of school adolescents [6]. The first 1,000 days represent a window of opportunity from conception through first 2 years after birth [7,8]. It represents a life window where nutritional deficiencies in the first 1000days can exert an overwhelming impacts [9–11].

It has been identified good nutrition and healthy growth during this period have lasting benefit throughout life by helping avert child mortality and lifelong disease burden, maximizing growth, enabling children to reach their cognitive and developmental potential,

particularly when combined with psychosocial stimulation [8,9,11–16]. However, it is not enough to know that a nutrition intervention is efficacious; it is also necessary to identify and address barriers and enablers for effective implementation of interventions under large-scale, real-world conditions.

To this effect, Addis Ababa University with the Federal Ministry of Health (FMOH) of Ethiopia and Irish Aid conducted an implementation research to develop four model nutrition districts. The overarching goal of the project was to create four model districts which implement evidence based, high impact and cost-effective package of nutrition interventions through the continuum of care; during delivery and through to infancy and up to 24 months old, from adolescent to pregnancy (motherhood). This study was conducted as part of the implementation research with the aim of assessing structural readiness of health facility and extent of provision of nutrition services at different contact points in four implementation districts.

## Materials and methods

### Study setting

The present study was part of a broader implementation research by Addis Ababa University in collaboration with the FMOH of Ethiopia and Irish Aid. The aims of the implementation research were; (i) Understanding barriers and facilitators of comprehensive and integrated nutrition service at primary health care unit (PHCU) level;(ii) Developing implementation protocol based on the finding from desk review, rapid assessment and barrier and facilitator analysis; (iii) Piloting the implementation protocol on selected kebeles; and (iv) Program learning and optimizing the implementation to create model nutrition districts.

This study was conducted with the aim of assessing the overall nutritional services provision in selected districts. It was conducted in four regions of Ethiopia namely Amhara, Oromia, Tigray and Southern Nations, Nationalities and Peoples' Region (SNNPR). The districts included were South Achefer, Kombolcha, Raya Azebo and Offa respectively. The estimated catchment population for these districts were 161,644, 189,945, 159,347 and 135,136 respectively. There were a total of 26 health centers and 84 health posts in the four selected districts. Of which, 23 health centers and 33 health posts were selected randomly for this study. The health centers were staffed by health officers, nurses, pharmacists, laboratory technicians and midwives while the health posts were staffed by health extension workers.

South Achefer district is located in West Gojjam Zone Administration, North-West, Ethiopia. Subsistence agriculture is the main economic activity, and the income of majority of the households depends on agriculture [17]. Kombolcha district is located in Eastern Hararghe Zone, Oromia National Regional State (ONRS) 542 km east of Addis Ababa. About 8% of the population professed Ethiopian Orthodox Christianity [18]. Crop-livestock farming activity is the main source of income and employment to the society, where crop production takes the principal part of income of the society [19]. Raya Azebo district is located in Tigray region 652 km away from Addis Ababa. Ninety percent of the district is considered "midland" (1500–2300 m) and 10% is considered "lowland" (< 1500 m) [20].

Offa district is one of 16 districts in Wolaita Zone and is located at 383 km south of Addis Ababa on the Sodo Gofa main road. The Offa district, altitude ranges 1200 around river Gogara in the south, 2800 m above sea level in the north [21].

### Study design and period

We used a mixed method approach employing a quantitative study followed by a qualitative study. This study was conducted between June-August, 2018.

## Sample size and sampling procedure

The sample size for this study was estimated using a sample size formula for a single proportion. The assumptions used to calculate the sample size were; (i) percentage of women who received nutritional counseling about exclusive breast feeding during postnatal care PNC (P) at 50%, (ii) a 95% confidence level and (iii) 5% margin of error. The calculated sample size was 384. We allocated the sample size proportionally across the four districts.

For an in-depth understanding about the overall nutrition service provision, we conducted a total of 60 key informant interviews in four districts. The key informants were district administration head ($n = 1$), district health office head ($n = 2$), deputy district health office head ($n = 2$), district nutrition focal person ($n = 4$), program officers ($n = 6$), service providers at health centers ($n = 24$), religious leader ($n = 1$), women development army (WDA) leader ($n = 1$), health extension workers (HEW) ($n = 9$) and PHCU directors ($n = 10$). We employed a purposeful sampling method to recruit key informants based on experience on nutrition service provision and management.

## Data collection tool and procedure

**Observation of services.**   The data collection addressed various dimensions of the comprehensive integrated nutrition services (CINS) provision at different contact points. The first dimension evaluated was the structural readiness. In this dimension, we assessed attributes of context in which care is delivered by observing availability of equipment, guidelines, social behavioral change communication (SBCC) materials and registration and reporting formats. Structural readiness was assessed in a total of 23 health centers and 33 health posts. We also assessed the characteristics of service providers including sex, age, level of education and nutrition training.

The second dimension evaluated was the process of service delivery. In this dimension, we observed the service provision against standard service guideline. This was done through direct observation of service providers while providing nutrition services at difference contact points. We observed contact points including Integrated Management of Neonatal and Childhood Illnesses (IMNCI) unit, immunization unit, antenatal care (ANC) unit and Post-Natal care (PNC) unit. We observed a total of 103 IMNCI, 73 immunization, 94 ANC and 79 PNC service provision. Some of the observed services include anthropometric assessment and interpretation, provision of IFA supplementation, provision of nutrition counselling and essential counselling skill of service providers. We used the CINS guide published by the FMOH to develop observation checklists (S1 Checklist). We also pretested the tools and made all the necessary modifications prior to data collection.

**Key informant interviews.**   Interview guides were developed with open-ended questions to have high degree of flexibility. Interview guides covered topics on multi-sectoral collaboration, integration of nutrition services, planning of nutrition services, registration and reporting of service provision, supportive supervision, human resource and motivation of service providers (S1 File).

The date, time and place of interview were arranged following what was most favorable and comfortable for the participants. Interviews were conducted in a quite office and privacy was ensured to enable participants to feel free and express their opinions. All interviews were audio taped and took about an hour on average. All data were collected by the project coordinators assigned in the four districts. All data were reviewed for completeness and consistency.

## Statistical analysis

**Quantitative analysis.**   We used Epi data version 3.1 for data entry and the statistical software package STATA version 14 for data cleaning and analysis. Descriptive statistics was run

on the different components of the nutritional services. Frequency tables and graphs were used to further describe the data set. Differences in service provider's counselling skill across contact points were reported using Pearson's chi-square ($X^2$) test. Statistical significance was set at $p < 0.05$.

**Qualitative data analysis.** To analyze the qualitative data, we initially developed field note template. Using this template, we wrote an expanded field note for each interview. Then we developed analysis matrix where the rows consist the subgroup selected for the study i.e. the respondents and the columns stands for the topic of study (i.e. the question asked). Using this matrix, we then extracted key findings from the field notes. Quotations were used to further illustrate the responses of the respondents on important issues.

## Ethics approval and consent to participate

Ethical clearance was obtained from the Research Ethical Committee of School of Public Health, Addis Ababa University. In addition, permission was obtained from relevant Federal and local government offices. Written informed consent was obtained from the participants after the necessary explanation about the purpose, procedures, benefits and risk of the study have been made. The respondent's right to refuse few or all of the questions was respected at all times. In addition, privacy of participants and confidentiality of the information obtained was kept.

## Results

### Structural readiness

**Availability of essential equipment at health facility.** The structural readiness of health centers for nutrition service delivery is presented in Table 1. As indicated, the majority of ANC consultancy room (21/23) were equipped with weighing scale, while only (17/23) and (14/23) IMNCI and immunization room were equipped with weighing scale respectively. Height scale was available in (16/23) IMNCI and ANC consultation room while only (8/23) immunization room were equipped with height scale.

The structural readiness of health post for nutrition service delivery is shown in Table 2. Most of the health posts had baby weighing scale (32/33), MUAC tape (33/33) and reporting formats (30/33). We also observed a critical shortage of equipment and guidelines and only few of the health posts had Adolescent, Maternal, Infant and Young Child Nutrition (AMIYCN) guideline (9/33), height board (8/33) and demonstration equipment (6/33).

**Table 1. Availability of equipment, guidelines and record keeping registers at health centers (n = 23).**

|  | Immunization | IMNCI | ANC | Delivery |
|---|:---:|:---:|:---:|:---:|
| **Anthropometric equipment** |  |  |  |  |
| Baby weighing scale | 18 | 18 | NA | 23 |
| Adult weighing scale | 14 | 17 | 21 | 14 |
| Height/length board | 8 | 16 | 16 | 0 |
| MUAC tape | 19 | 23 | 19 | 16 |
| **Counselling guide** |  |  |  |  |
| AMIYCN guideline | 10 | 10 | 8 | 8 |
| **Registration book** |  |  |  |  |
| Registration book and reporting format | 23 | 23 | 23 | 23 |

*NA not applicable

**Table 2. Availability of equipment, guidelines and record keeping registers at health posts (n = 33).**

| Equipment | Available and functional | Not available |
| --- | --- | --- |
| Baby weighing scale | 32 | 1 |
| MUAC tape | 33 | - |
| Reporting format | 30 | 3 |
| ANC registration book | 27 | 6 |
| Child card/record | 23 | 10 |
| Adult weighing scale | 20 | 13 |
| SAM guide line | 19 | 14 |
| PNC registration book | 15 | 18 |
| AMIYCN guide line | 9 | 24 |
| Height/length board | 8 | 25 |
| Demonstration equipment | 6 | 27 |

Table 3 shows availability of medical supplies for nutrition service in health centers and health posts. We observed that most of the health centers and health posts had adequate supply of drugs such as antibiotics, ORS, Iron folic acid, Zinc and Vitamin A. However, supplies such as Albendazole and RUTF were not available in nearly quarter of the health centers observed. It is important to note that Daily ration is nearly absent in all health centers and health posts. Only one health center and seven health posts had the daily ration at the time of observation.

**Nutrition training of health care providers.** As shown in Table 4, more than half (54.5%) of ANC service providers were clinical nurses/midwives and were predominantly female (80.7%). In addition, 53.1% of service provider in IMNCI were clinical nurses/midwives and were predominantly male (68.7%). Less than half of service providers in ANC (40.4%), PNC (40.4%) and IMNCI (42.2%) received training on nutrition.

## Process evaluation

**Anthropometric assessment.** We assessed the nutritional service delivery by observing a total of 103 IMNCI, 73 immunization, 94 ANC and 79 PNC service provisions. Fig 1 shows result from observation of anthropometric assessment at IMCNI service point. We found that, only 62 (60.2%) children had their weight measured, 38 (36.9%) had their height measured and 60 (58.3%) children had their MUAC measured. Of those children whose anthropometric

**Table 3. Availability of drug supply for nutrition services at health centers and health posts.**

| List of drugs | Health center (n = 23) | Health post (n = 33) |
| --- | --- | --- |
| Antibiotics (Amoxicillin, Cotrimoxazole) | 23 | 33 |
| ORS | 23 | 32 |
| Iron folic acid (60mg elemental iron + 0.4μg folic acid) | 23 | 31 |
| Zinc (20mg) | 23 | 30 |
| Vitamin A (50,000IU) | 21 | 30 |
| Albendazole (400mg) | 17 | 29 |
| RUSF | 16 | 20 |
| RUTF F-75 (75kcal/100ml) | 10 | NA |
| RUTF F-100 (100kcal/100ml) | 8 | NA |
| Daily rations (TSF) | 1 | 7 |

*NA not applicable

**Table 4. Characteristics of health care providers delivering ANC, PNC, immunization and IMNCI at health centers and health posts.**

|  | ANC (n = 57) | PNC (n = 52) | Immunization (n = 41) | IMNCI (n = 64) |
|---|---|---|---|---|
| **Sex of providers** |  |  |  |  |
| Male | 11 (19.3%) | 9 (17.3%) | 12 (29.3%) | 44 (68.7%) |
| Female | 46 (80.7%) | 43 (82.7%) | 29 (70.7%) | 20 (31.3%) |
| **Provider's Age (mean ±SD)** | 26.4 ± 3.68 | 27.3± 4.26 | 28.75± 5.05 | 28.2± 3.35 |
| **Level of education** |  |  |  |  |
| Bsc (Health officer, nurse and midwife | 15 (26.3%) | 9 (17.3%) | 4 (9.7%) | 22 (34.4%) |
| Diploma (Nurse and midwife | 31 (54.4%) | 30 (57.7%) | 18 (43.9%) | 34 (53.1%) |
| HEW | 11 (19.3%) | 13 (25%) | 19 (46.4%) | 8 (12.5%) |
| **Nutrition training** |  |  |  |  |
| Trained | 23 (40.4%) | 21 (40.4%) | 27 (65.8%) | 27 (42.2%) |
| Not trained | 34 (59.6%) | 31 (59.6%) | 14 (34.2%) | 37 (57.8%) |

measurement was taken, correct interpretation was conducted for only 32 (51.6%) children using the weight for age Z-score and for 28 (73.7%) children using height for age Z-score. Correct interpretation using MUAC cutoff point was conducted for only 39 (65%) children. At immunization service unit, only 12 children had their weight measured while none of the children had their height measured.

We observed a total of 94 ANC and 79 PNC service provision to assess the nutritional service delivery for mothers. We found that only 75 (79.8%) mothers had their weight measured and 41 (43.6%) had their MUAC measured at the ANC service point. In addition, only 15 (15.9%) mothers had their height measured. Of those mothers whose MUAC was measured, correct interpretation using MUAC cutoff was conducted for only 34 (82.9%) mothers.

**Nutrition counselling provision.** Table 5 shows nutrition counselling provided to mothers who came for IMNCI service. Out of 31 mothers with children under six months of age who visited IMNCI, only 19 (61.3%) were counselled on exclusive breastfeeding for the first six months. We also found that counselling on complementary feeding is suboptimal. For

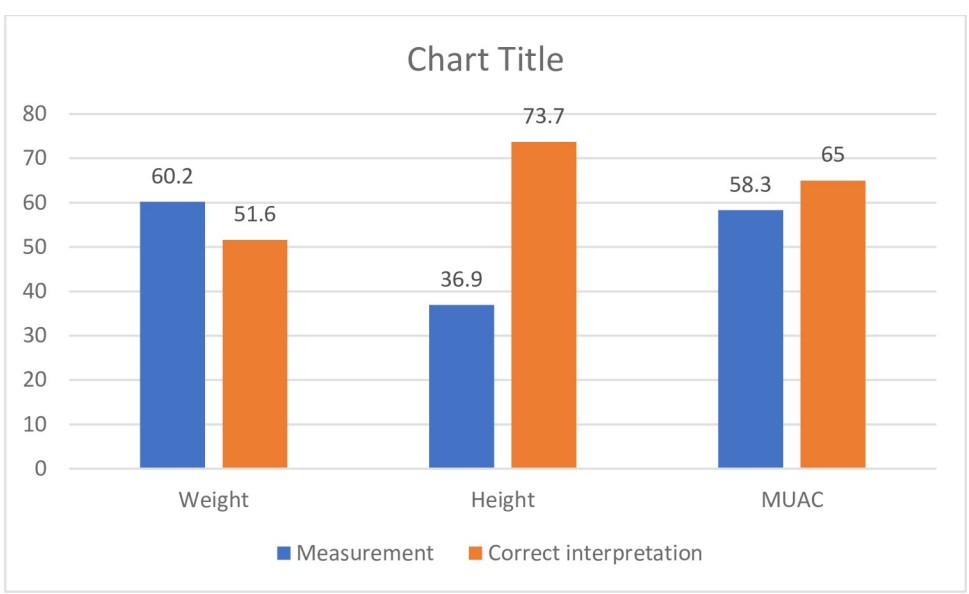

**Fig 1. Proportion of children receiving anthropometric measurement during IMNCI service provision.**

**Table 5. Proportion of mother receiving nutrition counselling at IMNCI service provision point.**

| Nutrition counselling during IMNCI service (n = 103) | Frequency (%) |
|---|---|
| **Counselling on breastfeeding** | |
| Advised on exclusive breastfeeding for the first 6 months (n = 31) | 19 (61.3%) |
| Advised on continuing frequent, on-demand breastfeeding until 2 years of age (n = 84) | 27(32.1%) |
| **Counselling on complementary feeding** | |
| Advised on initiation of complementary food at 6 months (n = 31) | 13(41.9%) |
| Advised on diversity to feed a child from grain, roots and tubers (n = 72) | 25(34.7%) |
| Advised on diversity to feed a child from legumes and nuts (n = 72) | 23(31.9%) |
| Advised on increasing fluid intake during illness, frequent breastfeeding, and encourage the child to eat soft, varied, appetizing, favorite foods (n = 103) | 26(25.2%) |
| Advised on diversity to feed a child from fruits and vegetables (n = 72) | 18(25.0%) |
| Advised on diversity to feed a child from animal source foods (n = 72) | 15(20.8%) |
| Advised on gradually increasing food consistency as the infant gets older (n = 72) | 16(22.2%) |
| Advised on the practice responsive feeding (n = 103) | 18(20.8%) |
| Advised on increasing the frequency of feeding with age(n = 72) | 15(17.5%) |
| **Counselling on WASH** | |
| Advised on personal and food hygiene (hand washing before food preparation, after using toilet, during feeding a child (n = 103) | 15(14.6%) |
| Advised on proper storage of food (n = 103) | 10(9.7%) |

example, of the total number of children illegible for counselling on initiation of complementary feeding, only 13 (42%) mothers were counselled about initiation of complementary food at six months. In addition, only 15 (20.8%) were advised on diversity i.e. to feed a child from animal source foods, and 15 (17.5%) were advised on increasing the frequency of feeding with age.

Table 6 shows the proportion of mothers receiving nutrition counselling at the immunization service point. We found that, out of 68 mothers with children under six months of age

**Table 6. Proportion of mothers receiving nutrition counselling at the immunization service provision point.**

| Nutrition counselling during Immunization service (n = 73) | Frequency (%) |
|---|---|
| **Counselling on breastfeeding** | |
| Advised on exclusive breastfeeding for the first 6 months (n = 68) | 11(16.2%) |
| Advised on continuing frequent, on-demand breastfeeding until 2 years of age (n = 73) | 5(6.9%) |
| **Counselling on complementary feeding** | |
| Advised on initiation of complementary food at 6 months (n = 68) | 9(13.2%) |
| Advised on increasing fluid intake during illness, frequent breastfeeding, and encourage the child to eat soft, varied, appetizing, favorite foods (n = 73) | 4(5.5%) |
| Advised on the practice responsive feeding (n = 73) | 3(4.1%) |
| Advised on increasing the frequency of feeding with age(n = 5) | |
| Advised on gradually increasing food consistency as the infant gets older (n = 5) | 0(0.0%) |
| Advised on diversity to feed a child from grain, roots and tubers (n = 5) | 0(0.0%) |
| Advised on diversity to feed a child from legumes and nuts (n = 5) | 0(0.0%) |
| Advised on diversity to feed a child from fruits and vegetables (n = 5) | 0(0.0%) |
| Advised on diversity to feed a child from animal source foods (n = 5) | 0(0.0%) |
| **Counselling on WASH** | |
| Advised on personal and food hygiene (hand washing before food preparation, after using toilet, during feeding of a child (n = 73) | 4(5.5%) |
| Advised on proper storage of food (n = 73) | 1(1.4%) |

who visited immunization unit, only 11 (16.2%) were counselled to exclusively breastfeed their child for six months and only 9 (13.2%) mothers were counselled about initiation of complementary feeding at 6 months. During the observation at immunization unit, we only had 5 mothers with children above the age of six months and no nutritional counselling was given for these mothers.

Table 7 shows various components of nutrition counselling provided to mothers who came for ANC follow-up. We observed suboptimal nutrition counselling service provided to pregnant mothers against the counselling standards set at ANC clinics. A total of 74 (78.7%) mothers were counselled on having one extra meal every day. While 46 (48.9%) mothers were counselled on having variety of food, 39 (41.5%) were counseled on the importance of having variety of food. Counselling on what to avoid such as eating raw food and what to limit such as caffeine was given for only for 4/94 and 3/94 mothers respectively. We have also observed that only 18 (19.2%) mothers were counselled on maternal health complications due to under nutrition during pregnancy.

Fig 2 shows counselling provided to mother on IFA supplementation during ANC follow up. We found that majority of the mothers 85 (90.4%) who came for ANC were provided IFA prescription. However, of those who were given prescription, only 73 (85.9%) were counselled to take IFA daily for six months. In addition, only 49 (57.7%) mothers were counselled about the benefit of IFA and 36 (42.4%) were counselled on the possible side effect of IFA.

Table 8 shows various components of nutrition counselling provided to mothers who came for PNC service. We have observed 75 (94.9%) mothers were counselled to exclusively breastfeed up to six months. Regarding maternal nutrition, 63 (79.8%) mothers were counseled to

**Table 7. Proportion of mothers receiving nutritional counselling at antenatal care service provision point.**

| Nutrition counselling (n = 94) | Frequency (%) |
|---|---|
| Advised the mother to have one extra meal/ snack every day | 74 (78.7%) |
| Advised the mother to eat from variety of food groups using locally available food | 46 (48.9%) |
| Advised the mother about the importance of eating variety of food groups | 39 (41.5%) |
| Advised the mother to have adequate fluid intake (*at least 8 glasses of fluid everyday*) | 29 (30.9%) |
| Advised the mother about fetal complications of under nutrition during pregnancy | 20 (21.3%) |
| Advised the mother about having seasonal fruits and vegetables everyday | 19 (20.2%) |
| Advised the mother about maternal complications of under nutrition | 18 (19.2%) |
| Advised the mother about the importance of exclusive breastfeeding for six month (4[th] visit) (n = 21) | 8 (38.1%) |
| Advised the mother on the importance of early initiation of breastfeeding (4[th] visit) (n = 21) | 4 (19%) |
| Advised the mother to avoid pre-lacteal feeding (4[th] visit) (n = 21) | 3 (14.3%) |
| Advised the mother about adding iodized salt when serving food (after cooking) | 13 (13.8%) |
| Advised the mother on importance of feeding colostrum (4[th] visit) (n = 21) | 1 (4.8%) |
| Advised the mother to expose the child to direct sun light (4[th] visit) (n = 21) | 1 (4.8%) |
| Advised the mother what to avoid during pregnancy (*i.e. Alcohol, raw/uncooked foods and vegetables*) | 4 (4.3%) |
| Advised the mother what to limit during pregnancy (*i.e. caffeine, fat*) | 3 (3.2%) |
| **Counselling on WASH (n = 94)** | |
| Advised the mother to keep personal hygiene | 46 (48.9%) |
| Advised the mother to wash hand after toilet use, and before meal preparation | 11 (11.7%) |
| Advised the mother to keep utensils and cooking material clean | 9 (9.6%) |
| Advised the mother to use treated water for drinking | 3 (3.2%) |
| **Counselling on ITN (n = 94)** | |
| Advised on ITN use | 34 (36.2%) |

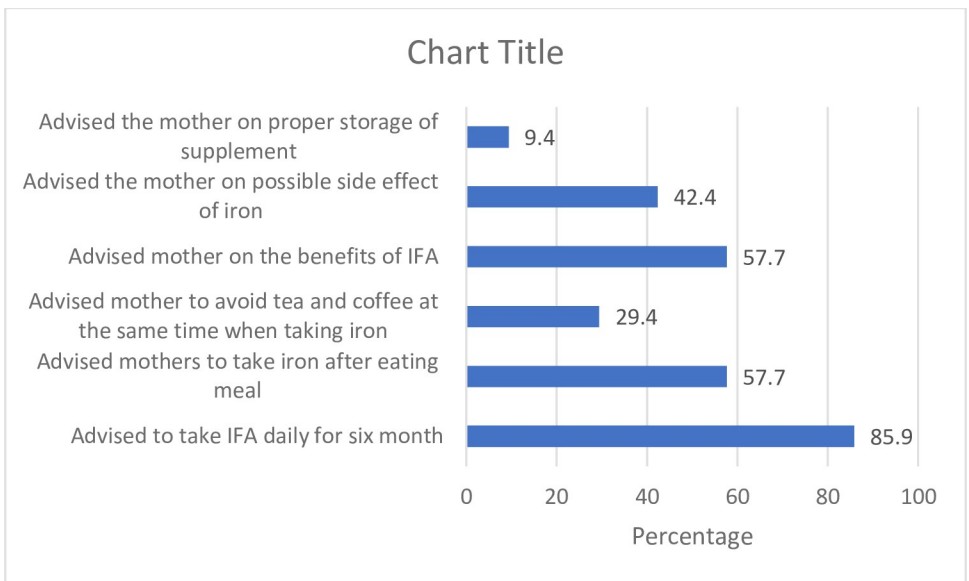

**Fig 2. Proportion of pregnant mother receiving counselling on Iron folic acid supplementation during antenatal care service provision.**

have two extra meal and rest during lactation. Counselling about maternal complication of undernutrition was given for only 27 (34.2%) mothers.

   **Counselling skill of service providers.**   Table 9 shows the counselling skill of service providers working at IMNCI and immunization service points. Listening to what the clients have

**Table 8. Proportion of mother receiving nutritional counselling at postnatal care service provision point.**

| Nutritional counselling (n = 79) | Frequency (%) |
|---|---|
| Advised mother to exclusively breastfeed up to six month | 75 (94.9%) |
| Advised the mother about extra meal and rest during lactation | 63 (79.8%) |
| Advised mother to breastfeed on demand | 62 (78.5%) |
| Advised the mother on the importance of exclusive breastfeeding for six month | 56 (70.9%) |
| Advised mother on correct positioning and attachment during breastfeeding | 54 (68.4%) |
| Advised the mother to eat from variety of food groups (locally available food) | 48 (60.8%) |
| Advised the mother to eat food rich in iron | 42 (53.2%) |
| Advised the mother to eat a diet that is rich in protein and fluid | 38 (48.1%) |
| Advised the mother on the importance of eating variety of food groups | 37 (46.8%) |
| Advised mother to avoid all dietary restriction if any | 35 (44.3%) |
| Advised the mother about maternal complications of under nutrition during lactation | 27 (34.2%) |
| Advised the mother to have adequate fluid intake (at least 8 glasses everyday) | 21 (26.6%) |
| Advised the mother to have seasonal fruits and vegetables everyday | 14 (17.7%) |
| Advised the mother about adding iodized salt when serving food (after cooking) | 13 (16.5%) |
| **Counselling on WASH (n = 79)** | |
| Advised the mother to use treated water for drinking | 15 (18.9%) |
| Advised the mother about safe disposal of potentially infectious soiled pads or other materials | 14 (17.7%) |
| Advised the mother to wash hand after toilet use, and before meal preparation, after cleaning baby, before feeding, before breast feeding | 12 (15.2%) |
| Advised the mother to keep utensils and cooking material clean | 15 (18.9%) |
| Advised the mother to keep personal hygiene | 40 (50.6%) |

**Table 9. Counselling skill of service providers during provision of counseling at IMNCI and immunization service points.**

| Counselling | IMNCI (n = 103) | Immunization (n = 73) | Chi2 P-value |
|---|---|---|---|
| Service provider listens to what the client has to say | 94 (91.3%) | 55 (75.3%) | **0.004** |
| Service provider recorded history of the client | 87 (84.5%) | 51 (69.9%) | **0.020** |
| Service provider used responses and gestures that show interest | 84 (81.6%) | 54 (73.9%) | 0.228 |
| Service provider avoided using judging words | 84 (81.6%) | 52 (71.2%) | 0.107 |
| Service provider Identified key difficulties (if any) and selects with the client the most important one to work on | 21 (20.4%) | 5 (6.8%) | **0.013** |
| Of those mothers with key difficulties service provider discussed options on key difficulties the client raised | 17 (80.9%)* | 5 (100%)* | 0.056 |
| Service provider recommends and negotiates do-able actions to help the client select the best option to try depending on her context and resources | 16 (76.2%)* | 5 (100%)* | 0.080 |
| Service provider helped the client agree to try one of the options and asks them to repeat the agreed-upon do-able action | 15 (71.4%)* | 5 (100%)* | 0.112 |
| Service provider let the client talk through her concerns before correcting information | 59 (57.3%) | 31 (42.5%) | 0.053 |
| Service provider recognized and praised what the client is doing correctly | 24 (23.3%) | 7 (9.6%) | **0.019** |

* The denominator is the number of clients identified with key difficulties

to say was higher among service providers at IMNCI 91.2% compared to service providers at immunization 75.3% *(P<0.05)*. Similarly, recognizing and praising what clients were doing correctly was higher among service providers at IMNCI 23.3% compared to service providers at immunization 9.6% *(P<0.05)*. During counselling, providers identified key difficulties for only 20.4% and 6.8% of the clients at IMNCI and immunization respectively *(P<0.05)*.

Table 10 shows the counselling skill of service providers working at ANC and PNC service points. Listening to what the clients have to say was higher among service providers at ANC 97.9% compared to service providers at PNC 89.9% *(P<0.05)*. Similarly, identifying key difficulties the clients have was higher among service providers at ANC 40.4% compared to service providers at PNC 16.5% *(P<0.001)*. However, discussing about options for the identified difficulties was higher among service providers at PNC 92.3% compared to service providers at

**Table 10. Counselling skill of service providers during provision of counseling at ANC and PNC service points.**

| Counselling | ANC (n = 94) | PNC (n = 79) | Chi2 P-value |
|---|---|---|---|
| Service provider listens to what the client has to say | 92 (97.9%) | 71 (89.9%) | **0.045** |
| Service provider used responses and gestures that show interest | 89 (94.7%) | 71 (89.9%) | 0.232 |
| Service provider let the client talk through her concerns before correcting information | 84 (89.4%) | 63 (79.8%) | 0.078 |
| Service provider avoided using judging words | 83 (88.3%) | 63 (79.8%) | 0.123 |
| Service provider Identified key difficulties (if any) and selects with the client the most important one to work on | 38 (40.4%) | 13 (16.5%) | **0.001** |
| Of those mothers with key difficulties service provider discussed options on key difficulties the client raised | 28 (73.7%)* | 12 (92.3%)* | **0.023** |
| Service provider recommends and negotiates do-able actions to help the client select the best option to try depending on her context and resources | 26 (68.4%)* | 10 (76.9) * | **0.015** |
| Service provider helped the client agree to try one of the options and asks them to repeat the agreed-upon do-able action | 26 (68.4%)* | 9 (75%)* | 0.491 |
| Service provider recognized and praised what the client is doing correctly | 43 (45.7%) | 25 (31.7%) | 0.059 |
| Service providers recorded history of the client | 71 (70.9%) | 56 (70.9%) | 0.083 |

* The denominator is the number of clients identified with key difficulties

ANC 73.7% *(P<0.05)*. Similarly, recommending doable action for the identified problem was higher among service providers at PNC 76.9% compared to service providers at ANC 68.4% *(P<0.05)*.

## Qualitative result

A total of 60 key informants participated in this study. The participants include district administrative head, district health office head, district health office deputy head, district nutrition focal person, program officers, service providers at health centers, religious leader, WDA, HEW and PHCU heads.

**Multi-sectoral collaboration.**   We found that the Woreda nutrition coordination body (WNCB), Woreda nutrition technical committee (WNTC) and Kebele nutrition technical committee (KNTC) are in place in the four districts. These structures were established following the 1000 days plus program launching. Key informant interview indicated that the issue is not related to the presence of the multi-sectoral coordination bodies rather it is related to its functionality. It is important to note that neither the coordination body nor the technical committee meet regularly and are productive enough.

One of the key informants explained about the functionality of WNCB, WNTC and KNTC as follows:

*"The WNCB meetings are not conducted regularly. There is usually third body request to have the meeting. It is not scheduled ahead and usually done hastily. There is unequal commitment of sectors, not planning together and there are no terms of reference. I can say WNTC almost does not exist and needs reestablishment. Regarding KNTC; the HEWs have capacity gap, they are not committed and motivated because of carrier structure problem, inadequate salary, no recognizing/rewarding mechanism, work overload etc. The kebele leaders are not committed as well."*

The functionality of the structure is mainly dependent up on the existence of nutrition projects in the district. All key informants believe that having WNCB is very important for the success of nutrition services. However, there is lack of ownership among different members.

One participant explained the issue of ownership as follows:

*"There is a tendency of pushing this job aside by the health sector due to different emerging activities and other priorities. In addition, the other sectors are dependent on an external initiation and they don't own this structure as part of their routine job. Apart from this there is also lack of logistic to conduct the meeting effectively. . .. you now like incentives."*

Another important bottleneck related to the functionality of the WNCB is the absence of shared definitive roles and responsibilities among the members. The participants of this study indicated that the members in the coordination body externalize their responsibility to the health sector and there is an assumption that nutrition is the concern of health sector only.

One of the participant expressed the following:

*"Because of other political priorities the coordination body is not operating to the envisioned extent. So mainly, it is the technical committee that is working better currently. However, there are certain still limitation on the technical committee as well. There is a high challenge with ownership of the body as they feel this task is solely the responsibility of the health sector".*

**Integration of nutrition services.**   Our assessment shows that there is sub optimal and poor integration of nutrition services with in child and maternal health services at the health centers and health posts. We documented a physical rather than functional integration of nutrition services. For instance, growth monitoring and promotion (GMP), immunization and IMNCI service are given in one room usually by one service provider. However, there is missed opportunity of providing GMP for children who came for immunization and IMNCI service.

*"There is no separate room for GMP, IMNCI and immunization at health center level. These three activities are offered in a room by a provider. The same is true for ANC, Delivery and PNC . . .. However; service providers are not using this opportunity to give nutrition service in an integrated manner."*

**Human resource, training and motivation of service providers.**   We have assessed staff adequacy, training and motivation of service providers at different levels in the districts. According to our finding, there is critical staff shortage at each level from district health office to the health post. This staff shortage was more profound at remote health facilities.

*"I myself focal for three positions, work in regular and emergency outpatient department (OPD), the health center head also works in OPDs, other staffs also have burden and generally there is staff shortage; there has been very critical shortage though yesterday one health officer joined the health center. There are only 5 staffs including the head and me. There is shortage of staffs in all departments."*

Even though there is shortage at health centers, it was more significant at health posts. The number of the HEWs was not comparable with the number of households living in their catchment area. One participant explained:

*"Based on the standard one health extension worker gives service to 2500 populations in her catchment area (1:2500). But the ratio in our woreda is 1: 10,000 to 11,000 populations. This is similar in the health center as well, there is not enough service provider compared to service seekers."*

Besides staff shortage, we documented high staff turnover. Similar to staff shortage, the turnover was more prominent in rural health facilities. The turnover was most common among midwives, health officers and HEW.

*"This year 8 HEWs left their job and moved to different cities by changing their job. It became a big challenge for the woreda. Some health posts have only one HEW. Currently Zemo health post have no HEW as a result it is closed."*

We have identified training gaps in AMIYCN, out-patient therapeutic program (OTP), community-based nutrition (CBN), management of sever acute malnutrition, blended integrated nutrition and counselling skill. Besides, the training opportunities mainly depend on the partner's or training provider organization's objective rather than the need of the health sector in the district. Furthermore, we have identified poor opportunities for health care providers to further their education and future career development.

*". . .there is no study opportunity. As I mentioned for you earlier, I joined this health center nearly five years back. It is very difficult to talk on this and there is no chance to pursue our education. Even when there are opportunities, there is no fair selection of candidates."*

We also evaluated motivation of service provider. We have found poor motivation among service providers. Participants described variety of factors affecting their motivation including access to training, access to further education, timely duty payment, work overload, and inadequate salary.

**Planning of nutrition services.**   The five-year strategic health sector plan is used as a baseline for district health office planning purpose. During planning HEWs, health center heads, personnel from district health office, and delegates from other sectors such as children and women affairs, agriculture, finance etc. are usually involved. However, the planning followed top to bottom approach. This approach does not take in to account the existing situation on the ground. Because of this, service providers had to deal with numbers sent from the top-level officials that doesn't really exist in their catchment area. They have explained this as follows:

*"We are facing challenge during planning as we breakdown the figures we took from the woreda to each health post by using conversion factors. Therefore, the plan is not based on the existing situation of the PHCU."*

*"There was a time where they give us 5000 plans while we have 3000 in our catchment and vice versa so don't get surprised when you see 205% achievement."*

**Registration and reporting of nutrition services.**   We observed multiple registration books being used by professionals for different services. Registration format were not available at facilities according to the standard set by the FMOH. We documented shortage of registration books especially at health post level.

*"We are using self-made registry. This is a burden for us since making the register is tiresome. It also has drawback as we might omit some important variables".*

Another challenge with registration book was related with portability. Health extension workers used a plain paper to document activities during outreach services because it is difficult to carry the registration books to outreach sites. This has drawback as the HEWs often forget to copy to the registers resulting in poor documentation and inconsistency among reports and registers. In general, there is poor documentation especially at health post level.

*"There is no documentation problem at health center level. We perform activities and record accordingly . . . . . . . . . but there is a gap at health post level. Last time the health center head and I along with kebele leader came across poor documentation. This might be because the HEW are newly assigned since senior HEWs have left their job for different reasons."*

Similar to registration, we have identified challenges related with reporting as well. The challenges were lack of timeliness, reliability, and completeness of the reports.

*"There was an incident where one of the health extension worker sent 0 ANC on the HMIS report and 20 ANC on the disease report, so we will cross check for such kind of errors and give feedback on how to improve the report in the future".*

**Supportive supervision.** The district health office provides supportive supervision to the health centers and health centers provide supportive supervision to health posts. There were two forms of supportive supervision in the district; integrated supportive supervision and program specific supportive supervision. As the name implies, integrated supportive supervision was conducted every quarter by integrating different case team leaders from the district health office. The program specific supervision was conducted whenever the need arises by individuals assigned to a specific program. This type of supervision focuses on specific programs.

*"There is Integrated Supportive Supervision (ISS) this will be done every quarter where different officers from the Woreda health office supervise health centers, health posts and at least 10 sample households at community level."*

## Discussion

The landscape analysis was intended to assess service readiness and nutrition service provision at different service provision points. The result from this study indicated poor structural readiness and gap in nutrition service provisions. Health facilities lacked essential anthropometric instruments, nutrition guidelines, registration and reporting formats. Poor structural readiness was more prominent at health posts compared to health centers. Our process evaluation indicated that there is missed opportunity of anthropometric assessment and preventive nutrition counselling at different contact points. Such problem was more critical at immunization unit than other units. In our qualitative assessment, we found that there is an existing but non-functional Woreda nutrition coordination body and technical committee. Training gaps, staff shortage and high staff turnover were the main challenges faced by service providers resulting in burden, fatigue and poor motivation.

We have observed poor structural readiness in both health centers and health posts. However, overall facility preparedness was better in health centers compared to health posts. This might be because the health centers provide comprehensive primary health care including promotive, curative and rehabilitative services as compared to health posts. Even though health centers exhibited better structural readiness, the level of readiness varied across different units. For instance, there were greater availability of weighing scale at ANC units than other units. Such type of problems were also common in other countries like Bangladesh where ANC rooms have better structural readiness than other units [22]. This might be because ANC has been practiced well as part of the health service implementation compared to the recently introduced nutrition service implementation in IMNCI [22].

Integration of nutrition service is expected at different contact points such as IMNCI. It is assumed that if screening is performed routinely in curative clinics, it creates a better opportunity for timely detection and intervention. However, our assessment of service provision at IMNCI indicated a missed opportunity for basic nutrition screening and counselling. Indeed, of the total number of children who came for IMNCI service, only 60.2% had their weight measured. It is important to note that the current IMCNI algorithm asks for weight of the child and yet 40% of the children were not weighed. Similarly, a study done in Botswana showed limited nutrition and dietary screening in curative clinics [23]. This missed opportunity could be attributed to lack of anthropometric equipment, poor motivation of staffs, shortage of staff and lack of supervision.

Poor interpretation of anthropometric assessments were also the findings from this study. Without nutrition assessment and correct interpretation, it is unlikely there would be

appropriate management of undernutrition and appropriate treatment of other childhood illness [24]. Of those children whose weight was measured at IMNCI unit, correct interpretation using weight for age Z-score was done for only 51.6% of the children. This might be related with capacity gaps, poor commitment and workload among service providers.

The other challenge that we identified was related to provision of preventive services such as proper counselling during clinical contact. Poor nutrition counselling could be a result of high caseload, staff shortage and capacity gap which led to prioritization of the immediate clinical concern rather than focusing on preventive intervention. However, overall nutrition service at IMNCI was relatively better as compared to immunization unit. We have identified a critical missed opportunity at immunization unit where only 16.4% of children had their weight measured while none of the children had their height measured. Lack of essential anthropometric instruments such as weight scale and stadiometer could have been possible reason for lack of appropriate nutritional assessment. In addition to lack of essential equipment, high caseload during immunization day could also be the possible reason for missed opportunity. The trend of provision of immunization was using a fixed day resulting in high caseload making immunization one of the most rushed services. This has resulted in poor nutrition counseling service at immunization unit. Of all the services, the least preventive counselling service was observed in this unit.

Immunization services are often widely available and potentially can support and be supported by additional health interventions. The combined delivery or integration of linked health interventions including GMP, nutrition advises and preventive cares for a more effective way of achieving common health goals [25]. In the study area, there was an opportunity to integrate nutrition services as there was a physical integration of IMNCI, Immunization and GMP services. These services were given in a single room and by a single provider in most of the health facilities. However, the service provision did not progress beyond the physical integration which resulted in missed opportunities.

The level of provision of essential nutrition services was better at ANC and PNC unit as compared to IMNCI and immunization unit. This is reflected by improved nutrition counselling at ANC and PNC unit. Nutritional assessment was also better at ANC unit. For instance, majority (79.79%) of the mothers at ANC were weighed. We have also observed a relatively better counselling at ANC unit. This improved nutrition service might be due to the preventive nature of ANC service. Since ANC service is preventive than curative it makes it conducive to provide nutrition counselling than other services.

Our study showed prescription of IFA was high however; counselling related to IFA was poor. This finding was similar with a report conducted with the aim of describing and comparing nutrition intervention using service readiness and delivery across different countries. The finding from the study indicated prescription/provision of IFA was higher than counselling service related with IFA. Prescription ranged from 62.2% in Nepal to 87.5% in Senegal. Nonetheless, counselling on benefit of IFA ranged from 18% in Nepal to 72.6% in Tanzania [26]. Counselling on possible side effects was very low ranging from 1.8% in Senegal to 14.2% in Tanzania [26]. Inadequate tools and skill in counselling to support and monitor adherence could be possible reason for the observed result [27]. Poor counselling service about benefits, possible side effects, severity and magnitude of the problem has been identified as barriers for non-adherence of IFA [26,28–30]. A systematic review has identified fear of side effect as the commonest reason for non-adherence which was 46.4% [31]. As a result, counselling addressing benefit and possible side effect with mitigation strategy to overcome the side effect should be incorporated other than mere provision of the supplement to ensure adherence.

During our assessment, we have identified major challenges faced by service providers. Training gap, staff shortage and high staff turnover were the commonly identified challenges. The

training gap was not uniform in all unit; it was more pronounced among health care providers at maternity unit. The other challenge with training is that it was not need based rather it mainly depend on the objective of the organization providing the training. Similar to training, staff shortage and turnover was also not uniform. It was more common at rural health facilities. This might be due personal reasons and service provider's desire to live in town or near to town. Such problems could be possible reasons for poor implementation of services at different unit.

Motivation of staff was one factor for effective delivery of nutrition services. Health extension workers are overloaded with health and other non-health related commitments such as political tasks. This has resulted in poor motivation and fatigue among health extension workers. Apart from the burden of work, lack of access to trainings, lack of timely duty payment, lack of advancement in national career structure and inadequate salary were factors for poor motivation of service providers.

Because of the multi-sectoral nature of nutrition, various stakeholders outside the health needs to be involved, coordinated and linked to gain the synergistic impact for achievement of nutrition interventions [6]. Even though nutrition coordinating bodies and technical committees were established in all the districts, it is not functioning to the desired level. The primary reason for this could be engagement of different sectors in other urgent and political commitments. Even though the sectors will be housed in health sector all member of the coordinating body and technical committee should be equally accountable [32]. However, this was not the reality in all districts. There was poor ownership and accountability among sectors reflected by externalizing every responsibility to the health sector. This might be a result of lack of strong monitoring and evaluation system in place. In addition, lack of coordination from the federal and regional governments has been identified as a constraint to improve nutrition service delivery in Ethiopia [33]. In order to ensure functionality of coordination body and technical committee, there should be a mechanism to capture data and triangulate from all relevant sectors.

We believe the current study has limitation that needs to be acknowledged. Since we used direct observation, it is difficult to rule out desirability bias. Direct observation during service provision may have encouraged service providers to offer a more thorough service for their clients. However, we believe the impact is minimal since our finding on both the structural readiness and process evaluation were low. The strength of this study was the assessment of both health centers and health posts and the use of multiple methods of data collection techniques including observation, and key informant interviews to assess the structural readiness and provision of nutrition services.

## Conclusion and recommendation

We documented poor structural readiness and gaps to deliver integrated nutrition services in all four districts. There were significant missed opportunities for different nutrition services including nutrition screening and counselling. Poor commitment and accountability of coordination body and technical committee towards nutrition intervention was common. Improving logistics and training may improve delivery of nutrition service. Improving health human resource might reduce missed opportunity that resulted because of high caseload. It might also ensure adequate time to provide thorough preventive counselling service at all units. Ensuring a clear structure of accountability along with reporting mechanisms in the existing system might ensure functionality and accountability of the coordinating body and technical committee.

## Supporting information

**S1 Dataset. The minimal data set of this study.**
(XLSX)

**S1 Checklist. Observation checklist used in this study.**
(DOCX)

**S1 File. Key informant interview guide used in this study.**
(DOCX)

## Acknowledgments

We would like to thank Addis Ababa University, School of Public Health for facilitating the conduct of the research. We appreciate the managers and health care providers for their assistance and cooperation during the assessment. Finally, we would like to thank all study participants who took part in this study.

## Author Contributions

**Conceptualization:** Esete Habtemariam Fenta, Bilal Shikur Endris, Yalemwork Getenet Mengistu, Fekadu Elias Sadamo, Endashaw Hailu Gelan, Tsegaye Gebrezgher Beyene, Seifu Hagos Gebreyesus.

**Data curation:** Esete Habtemariam Fenta, Bilal Shikur Endris, Yalemwork Getenet Mengistu, Fekadu Elias Sadamo, Endashaw Hailu Gelan, Tsegaye Gebrezgher Beyene, Seifu Hagos Gebreyesus.

**Formal analysis:** Esete Habtemariam Fenta, Bilal Shikur Endris, Yalemwork Getenet Mengistu, Fekadu Elias Sadamo, Endashaw Hailu Gelan, Tsegaye Gebrezgher Beyene, Seifu Hagos Gebreyesus.

**Funding acquisition:** Bilal Shikur Endris, Seifu Hagos Gebreyesus.

**Investigation:** Esete Habtemariam Fenta, Bilal Shikur Endris, Yalemwork Getenet Mengistu, Fekadu Elias Sadamo, Endashaw Hailu Gelan, Tsegaye Gebrezgher Beyene, Seifu Hagos Gebreyesus.

**Methodology:** Esete Habtemariam Fenta, Bilal Shikur Endris, Yalemwork Getenet Mengistu, Fekadu Elias Sadamo, Endashaw Hailu Gelan, Tsegaye Gebrezgher Beyene, Seifu Hagos Gebreyesus.

**Project administration:** Esete Habtemariam Fenta, Bilal Shikur Endris, Yalemwork Getenet Mengistu, Fekadu Elias Sadamo, Endashaw Hailu Gelan, Tsegaye Gebrezgher Beyene, Seifu Hagos Gebreyesus.

**Resources:** Bilal Shikur Endris, Seifu Hagos Gebreyesus.

**Software:** Esete Habtemariam Fenta, Bilal Shikur Endris, Yalemwork Getenet Mengistu, Fekadu Elias Sadamo, Endashaw Hailu Gelan, Tsegaye Gebrezgher Beyene, Seifu Hagos Gebreyesus.

**Supervision:** Esete Habtemariam Fenta, Bilal Shikur Endris, Yalemwork Getenet Mengistu, Fekadu Elias Sadamo, Endashaw Hailu Gelan, Tsegaye Gebrezgher Beyene, Seifu Hagos Gebreyesus.

**Validation:** Esete Habtemariam Fenta, Bilal Shikur Endris, Yalemwork Getenet Mengistu, Fekadu Elias Sadamo, Endashaw Hailu Gelan, Tsegaye Gebrezgher Beyene, Seifu Hagos Gebreyesus.

**Visualization:** Esete Habtemariam Fenta, Bilal Shikur Endris, Yalemwork Getenet Mengistu, Fekadu Elias Sadamo, Endashaw Hailu Gelan, Tsegaye Gebrezgher Beyene, Seifu Hagos Gebreyesus.

**Writing – original draft:** Esete Habtemariam Fenta, Bilal Shikur Endris, Yalemwork Getenet Mengistu, Fekadu Elias Sadamo, Endashaw Hailu Gelan, Tsegaye Gebrezgher Beyene, Seifu Hagos Gebreyesus.

**Writing – review & editing:** Esete Habtemariam Fenta, Bilal Shikur Endris, Yalemwork Getenet Mengistu, Seifu Hagos Gebreyesus.

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
