## [Decision Letter · Decision Letter 0]

4 Sep 2020

PONE-D-20-17292

Landscape analysis of nutrition services in primary health care unit, Ethiopia.

PLOS ONE

Dear Dr. Fenta,

Thank you for submitting your manuscript to PLOS ONE. After careful consideration, we feel that it has merit but does not fully meet PLOS ONE’s publication criteria as it currently stands. Therefore, we invite you to submit a revised version of the manuscript that addresses the points raised during the review process.

We look forward to receiving your revised manuscript.

Kind regards,

Jianhong Zhou

Associate Editor

PLOS ONE

Journal Requirements:

2. Please include additional information regarding the survey or questionnaire used in the study and ensure that you have provided sufficient details that others could replicate the analyses. For instance, if you developed a questionnaire as part of this study and it is not under a copyright more restrictive than CC-BY, please include a copy, in both the original language and English, as Supporting Information

Reviewers' comments:

Reviewer's Responses to Questions

**Comments to the Author**

1. Is the manuscript technically sound, and do the data support the conclusions?

Reviewer #1: Partly

Reviewer #2: Partly

2. Has the statistical analysis been performed appropriately and rigorously? 

Reviewer #1: No

Reviewer #2: Yes

3. Have the authors made all data underlying the findings in their manuscript fully available?

Reviewer #1: Yes

Reviewer #2: Yes

4. Is the manuscript presented in an intelligible fashion and written in standard English?

Reviewer #1: Yes

Reviewer #2: Yes

5. Review Comments to the Author

Reviewer #1: Excellent topic and conceptualization of the problem being addressed by this research. Methods and topics investigated are aligned with the purpose of answering key questions for this research. In particular, details of the assessment on counseling quality are valuable to arrive at conclusions about process of service delivery and adequacy of front line service provider skills. However, more statistical analysis of differences between PHC and health posts, and across ANC, immunization and IMNCI are needed to support the conclusions. Finally, a more direct and clear link between district level coordination and technical oversight and the observed status of structural readiness and processes would be helpful.

Reviewer #2: I have included my comments to the authors in the Word file of the manuscript and uploaded as Reviewer's attachment.

I have included my comments to the authors in the Word file of the manuscript and uploaded as Reviewer's attachment.

6. PLOS authors have the option to publish the peer review history of their article (what does this mean?). If published, this will include your full peer review and any attached files.

Reviewer #1: No

Reviewer #2: No

---

## [Author Response · Author response to Decision Letter 0]

13 Oct 2020

Reviewer comments:

Reviewer 1: 

1. Excellent topic and conceptualization of the problem being addressed by this research. Methods and topics investigated are aligned with the purpose of answering key questions for this research. In particular, details of the assessment on counseling quality are valuable to arrive at conclusions about process of service delivery and adequacy of front line service provider skills. However, more statistical analysis of differences between PHC and health posts, and across ANC, immunization and IMNCI are needed to support the conclusions.

• We have conducted chi square test to identify difference between counselling skill of service providers at ANC, PNC, immunization and IMNCI. We have presented the finding from the analysis in table 9 and 10 from line 338-359. However, as the number of observation from health centers is much higher than the number of observations from health posts We do not have adequate power to compare findings from health center with findings from health posts as the number of health posts included in this study are few compared to health centers

2. Finally, a more direct and clear link between district level coordination and technical oversight and the observed status of structural readiness and processes would be helpful.

• Even though this is an important point, as indicated in our qualitative findings, the coordination body and technical committees are not functioning to the level of expectation in all districts. Therefore, it is difficult for us to measure weather there is a direct link between coordination and technical oversight and structural readiness and process.

Reviewer 2: 

1. A suggested revised title “Landscape analysis of nutrition services in primary health care units of four districts in Ethiopia”

• We have changed the title to “Landscape analysis of nutrition services at Primary Health Care Units (PHCUs) in four districts of Ethiopia.” according to the suggestion given by the reviewer. 

2. I suggest including a sentence or two on recommendations.

• We have included recommendation in the abstract section from line 56-58. 

3. More description of the study setting, and study population should be useful. For example, nutritional status of populations groups, socioeconomic status, household food security, main livelihood/profession, local productions etc. 

• We have included description about the study setting from line 119-132. 

4. Results section is too long. For most of the tables, all data presented in the tables are also narrated as text. If only significant data presented in the tables are highlighted in the text and the tables are referred to – the length of the results section could be significantly reduced.

• We have tried to take some of the narrative from the result section and only focused on results that are significant. For instance we took out the result narration from line from 209-213, from line 223-225, from 269-270, from 280-281 and from line 314-315. 

5. Line 178-179 should be revised 

• We have revised the statement in line 206-207

6. Table title is not complete. No need to add Structural readiness of Health Centers to all titles. A suggested revised title – Availability of equipment, guidelines and record keeping registers at health centers

• We have revised all table titles to be complete 

7. The dose of all medicines and composition of nutrition supplements should be mentioned as much as possible.

• We have included doses of the medications in table 3. 

8. Fig 1 - I suggest excluding data on anthropometry at the immunization centers. These data are insignificant and have been mentioned as text. Otherwise, it is confusing that data from two places are presented but the correct interpretations are presented only for IMNCI. Again, text related to Fig 1 can be reduced.

• We have revised figure 1 to include only findings of anthropometric assessment and interpretation at IMNCI contact point only and described the finding from immunization in text only. 

9. Here and later where quotes are used, why is this needed. Shouldn’t it be kept confidential? I suggest taking this out from all quotes.

• We took out the age, sex, profession and district from all qualitative findings. 

10. Line 308 All or some? 

• All participants believe the importance of having woreda nutrition coordination body. Therefore, we have corrected the statement to indicate that all participants believe in the importance of WNCB in line 382. 

11. Maybe I am missing something, but I am not sure I understand this. What finding shows this? If this was explicit from the KIIs, then it should be mentioned.

• The finding from both quantitative and qualitative study indicates poor integration of nutrition services. In the quantitative part, poor integration is reflected in the finding that 40% of children who visited IMNCI went without their weight being measured and only 12 children out of 73 children who went for immunization had their weight measured. The same is true for counselling. Even though integrating nutrition services such as anthropometric assessment and nutrition counselling is expected at different units in health facilities we have observed there are clients that are not provided with such services. It is because of such findings that we said there is poor integration of nutrition services. In addition to our quantitative finding our qualitative finding is also indicative of poor integration of nutrition services in the health facilities presented from line 403-412. 

12. Line 309 Of what? Better to use self-explanatory headings/sub-headings.

• We have edited the title of headings/sub-heading to be self-explanatory in line 412, 448 and 463. 

13. This is not true as this was stated by only one KI who is 26 years, male, health officer.

• Two key informant explained the top to bottom approach in planning. One is a 26 years key informant explained that they use the figure they take from the woreda using conversion factor in line 456-458. The other key informant was a 28 years old nurse who explained the top to bottom approach by stating the numbers they get is higher than the actual number in their catmint are in line 460-461. In addition, we selected quotation that best describe the findings. 

14. Line 448 Equipment? Trained staff? Or something else?

• In line 526 we have included equipment 

15. I think this is an important aspect of this study. Proper anthropometric assessment and interpretation are not possible unless the staff responsible for this are well trained. There are some results highlighted from the qualitative study but nothing from the quantitative study. I suggest including results on training from quantitative study if that was captured. 

• We have assessed access to training by asking service providers whether they have received nutrition training. We have included this in the result section under the heading structural readiness and sub heading “Nutrition training of health care providers” from line 244-251. 

16. This is somehow repetitive and may be avoided. Also, this paragraph and the following one can be made as a single paragraph on IFA. 

• We have edited the statement from line 561-565.

17. References should be used.

• We have cited a reference in line 569

18. How was it assessed? If done, this should be mentioned in the Methods section. If not done, it should be included as a limitation of this study

• We have assessed access to nutrition training by asking service providers whether they have received Nutrition training. We have included this in the method section in line 159-160. 

19. What does this mean? Was this assessed? On what basis was the rate high?

• When we say missed opportunity, we are referring to the number of clients who visited the health facility for one of the services but who missed the opportunity to be screened for malnutrition and to be provided nutritional counselling. For instance, as indicated in our finding, IMNCI and immunization services provide an opportunity for the child to be screened and to be provided with nutrition counselling. However not every children and mother were able to get the screening and nutrition counselling. The same goes for ANC and PNC services. As for the question regarding high rate, We have removed the term rate in the statement in line 614 to clear ambiguity. 

20. I think these terminologies are used here very loosely. I think these were not explicitly measured in this study.

• We have measured this in the qualitative section and presented findings related with this under the heading multi-sectoral collaboration from line 293-232. In this part we have tried to assess existence of coordination body and technical committee, their functionality and so on. Our finding shows that there is existence of the body but their functionality is not to the level of expectation, in addition their functionality mainly depend on the existence of different nutrition sproject in the area. This shows that there is no system that is in place to the coordination body and technical committee accountable even if they are not functioning. In addition, different sectors externalize their responsibility to the health sector and this shows poor commitment. It is because of such findings that we reached into the conclusion that there is poor commitment and accountability among the coordination body and technical committee. 

21. Importance of training is mentioned here but no results were presented on this. See comment for Line 449.

• We have included access to training in the quantitative result as well from line 244-251. .

---

## [Editor Report · Decision Letter 1]

12 Nov 2020

PONE-D-20-17292R1

Landscape analysis of nutrition services at Primary Health Care Units (PHCUs) in four districts of Ethiopia.

PLOS ONE

Dear Dr. Fenta,

Thank you for submitting a revised manuscript titled "Landscape analysis of nutrition services at Primary Health Care Units (PHCUs) in four districts of Ethiopia". I participated as a reviewer for the initial evaluation of this manuscript. I have reviewed the revised manuscript against the reviewer's comments on the earlier version. In my view, all reviewer's comments have been addressed properly in the current version, thank you. However, the manuscript, in my view, will be benefitted if you could address the minor editorial issues throughout the manuscript. Below are a few examples but it would be good if a thorough editing can be done throughout the manuscript. 

ln 37: 'sixty' should be '60'; Ln 120-130: latitude/altitude information may be omitted; Ln 216 and 218: 'equipment’s' should be 'equipment'; Ln 225: 'RUTF’s' should be 'RUTF'; Ln 228: Table 3, row 1 - add a few examples of antibiotics; Ln 265 and 274: Table 5 and 6, row 1 - 'Exclusively' should be 'exclusive', row 2 - 'Continuing' should be 'continuing'.

We look forward to receiving your revised manuscript.

Kind regards,

Kuntal K. Saha, PhD.

Academic Editor

PLOS ONE

---

## [Author Response · Author response to Decision Letter 1]

13 Nov 2020

November 13, 2020

Dear Editor

PLOS ONE Journal 

we would like to thank you for reviewing the manuscript and giving us constructive comments and recommendations. We appreciate the opportunity given to us to revise and resubmit this manuscript. Below are our responses to the comments from the reviewers and a description of what changes we have made (texts in bullet points and italic font). 

Reviewer comments:

Reviewer 1: 

1. Thank you for submitting a revised manuscript titled "Landscape analysis of nutrition services at Primary Health Care Units (PHCUs) in four districts of Ethiopia". I participated as a reviewer for the initial evaluation of this manuscript. I have reviewed the revised manuscript against the reviewer's comments on the earlier version. In my view, all reviewer's comments have been addressed properly in the current version, thank you. However, the manuscript, in my view, will be benefitted if you could address the minor editorial issues throughout the manuscript. Below are a few examples but it would be good if a thorough editing can be done throughout the manuscript. ln 37: 'sixty' should be '60'; Ln 120-130: latitude/altitude information may be omitted; Ln 216 and 218: 'equipment’s' should be 'equipment'; Ln 225: 'RUTF’s' should be 'RUTF'; Ln 228: Table 3, row 1 - add a few examples of antibiotics; Ln 265 and 274: Table 5 and 6, row 1 - 'Exclusively' should be 'exclusive', row 2 - 'Continuing' should be 'continuing'

• We have made editorials in line 60, 216, 218, 225 accordingly. We omitted the latitude/altitude information in line 120-130. We have added example of antibiotics in table 3. We have also edited the comments raised in table 5 and 6. In addition, we have made editorials throughout the manuscript.

---

## [Editor Report · Decision Letter 2]

18 Nov 2020

Landscape analysis of nutrition services at Primary Health Care Units (PHCUs) in four districts of Ethiopia.

PONE-D-20-17292R2

Dear Dr. Fenta,

We’re pleased to inform you that your manuscript has been judged scientifically suitable for publication and will be formally accepted for publication once it meets all outstanding technical requirements.

Kind regards,

Kuntal K. Saha, PhD.

Guest Editor

PLOS ONE
---

## [Editor Report · Acceptance letter]

23 Nov 2020

PONE-D-20-17292R2 

Landscape analysis of nutrition services at Primary Health Care Units (PHCUs) in four districts of Ethiopia. 

Dear Dr. Fenta:

I'm pleased to inform you that your manuscript has been deemed suitable for publication in PLOS ONE. Congratulations! Your manuscript is now with our production department. 

Kind regards, 

on behalf of

Dr. Kuntal K. Saha 

Guest Editor

PLOS ONE